

# Ramification has little impact on shoot hydraulic efficiency in the sexually dimorphic genus *Leucadendron* (Proteaceae)

Adam B. Roddy[1], Justin J. van Blerk[2], Jeremy J. Midgley[2] and Adam G. West[2]

[1] Department of Integrative Biology, University of California, Berkeley, Berkeley, CA, United States of America
[2] Biological Sciences, University of Cape Town, Cape Town, South Africa

## ABSTRACT

Despite the diversity of branching architectures in plants, the impact of this morphological variation on hydraulic efficiency has been poorly studied. Branch junctions are commonly thought to be points of high hydraulic resistance, but adjustments in leaf area or xylem conduit abundance or dimensions could compensate for the additional hydraulic resistance of nodal junctions at the level of the entire shoot. Here we used the sexually dimorphic genus *Leucadendron* (Proteaceae) to test whether variation in branch ramification impacts shoot hydraulic efficiency. We found that branch ramification was related to leaf traits via Corner's rules such that more highly ramified shoots had smaller leaves, but that branch ramification had little consistent impact on shoot hydraulic efficiency, whether measured on a leaf area or stem cross-sectional area basis. These results suggest that the presumed increase in resistance associated with branching nodes can be compensated by other adjustments at the shoot level (e.g. leaf area adjustments, increased ramification to add additional branches in parallel rather than in series) that maintain hydraulic efficiency at the level of the entire shoot. Despite large morphological differences between males and females in the genus *Leucadendron*, which are due to differences in pollination and reproduction between the sexes, the physiological differences between males and females are minimal.

## INTRODUCTION

Plants are remarkably diverse in gross morphology. Theoretical studies have shown that the number of optimal plant architectures increases with the number of functions that must be performed (*Niklas, 1994*). One notable trait that varies both within and among species is the degree of branch ramification. Because branches and stems serve both hydraulic and biomechanical functions, larger leaves and leaf areas are correlated with thicker branches (*Corner, 1949*; *Ackerly & Donoghue, 1998*; *Olson, Aguirre-Hernández & Rosell, 2009*). According to Corner's rules, branch size is inversely related to how highly ramified shoots are; that is, given a constant allocation of biomass, the stems of more highly branched shoots should, on average, be smaller. Furthermore, branch size and

Corresponding author
Adam B. Roddy,
adam.roddy@berkeley.edu

ramification may also be linked to reproductive characters: in species that bear flowers on terminal shoots, selection on inflorescence size and number is associated with leaf size via the allometric scaling of each of them with stem size (*Bond & Midgley, 1988*; *Midgley & Bond, 1989*). Corner's rules has, therefore, provided a useful framework for studying plant architecture (*Lauri, 2018*). Subsequent theory sought to explain constancy in the relationship between leaf area and stem size with a hydraulic argument (*Shinozaki et al., 1964*), although the underlying assumption of the pipe model theory that all xylem in the stem are conductive is not valid (*Lehnebach et al., 2018*). One open question is, therefore, whether variation in stem size and branch ramification affects hydraulic efficiency: are more highly ramified shoots with smaller branches less hydraulically efficient? If more highly ramified branches are less hydraulically efficient, then branching would limit plant architectural variability and potentially trade-off with production of terminal structures (e.g., leaves and inflorescences).

Studies that have mapped the hydraulic architecture of trees have found large variability in leaf-specific stem hydraulic conductivity (i.e., the hydraulic conductivity normalized to the supplied leaf area) throughout the crowns of adult trees (*Tyree et al., 1983*; *Ewers & Zimmermann, 1984b*; *Ewers & Zimmermann, 1984a*; *Tyree & Ewers, 1991*; *Tyree & Alexander, 1993*; *Tyree & Zimmermann, 2002*). Stem segments immediately basal to branch junctions tend to have higher conductivity than the junctions themselves (*Ewers & Zimmermann, 1984a*; *Ewers & Zimmermann, 1984b*), although this difference is not as great as the variation in leaf specific conductivity throughout the rest of the crown (*Tyree & Alexander, 1993*). Nonetheless, branch junctions are typically thought to increase hydraulic resistance (*Tyree & Zimmermann, 2002*), meaning that more highly branched shoots may be less conductive. However, because most stem hydraulic measurements are made on unbranching stem segments, they typically include multiple years of growth and do not include the terminal branches where conduits taper dramatically (*Anfodillo, Petit & Crivellaro, 2013*; *Olson et al., 2014*). Furthermore, how conductivity of a stem segment (or loss of conductivity due to embolism in the stem segment) influences the hydraulic conductance of the entire shoot network is not immediately intuitive. How resistance (the reciprocal of conductance) is partitioned throughout the plant will dictate how conductivity of a stem segment scales with shoot conductance (*Meinzer, 2002*; *Brodersen et al., 2019*). The implication is that there can be significant declines in conductivity of a stem segment (due to either embolism formation or to branch junctions) without impacting the hydraulic conductance of the entire shoot.

Here we tested whether branch ramification reduces shoot hydraulic conductance. We compared males and females of the genus *Leucadendron* (Proteaceae). *Leucadendron* is known for its high variability in sexual dimorphism, with males and females of some species being highly dimorphic while males and females of other species exhibit little dimorphism. *Leucadendron* species display a full range of phenotypes—from monomorphism to dimorphism—in leaf size, branch size and ramification, and inflorescence size, with males generally having smaller, more numerous leaves and branches (*Williams, 1972*; *Bond & Midgley, 1988*; *Midgley & Bond, 1989*). The most striking differences between males and females of dimorphic *Leucadendron* species are differences in leaf size and

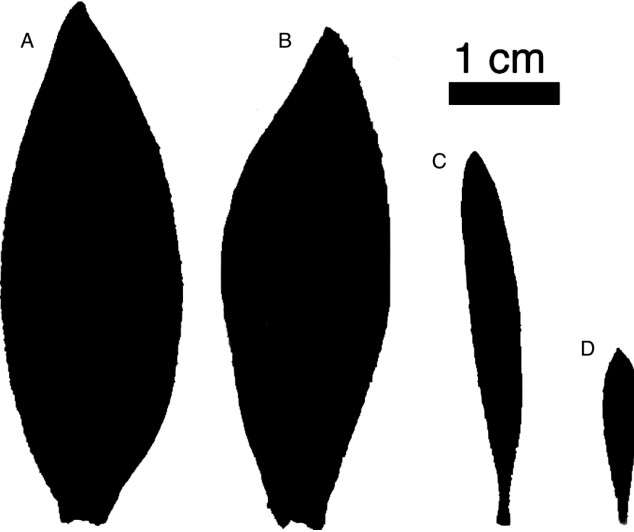

**Figure 1** **Representative leaves highlighting sexual dimorphism.** Representative leaves with areas approximating the average leaf area for each group. *L. daphnoides* (A) female and (B) male, *L. rubrum* (C) female and (D) male.

branch ramification (Fig. 1). Theoretical and empirical studies in other systems have long considered leaf size to be mechanistically linked to branch size because of the biomechanical constraints on the distribution of leaves (*Corner, 1949*; *Ackerly & Donoghue, 1998*; *Olson, Aguirre-Hernández & Rosell, 2009*). In dimorphic *Leucadendron* species, females have larger leaves and thicker, less ramified stems than their conspecific males. Furthermore, females of many *Leucadendron* species maintain their cones in the canopy for several years, a strategy termed 'serotiny'. Serotiny allows seeds to be dispersed only after fire, but preventing the cones from opening during the hot, dry Mediterranean summers requires a continuous supply of water and carbon. Although the water costs of maintaining cones are lower than originally thought (*Cramer & Midgley, 2009*), the relative costs of reproduction are assumed to be higher for females than for males (*Barrett & Hough, 2013* but see *Bond & Maze, 1999*), and these costs are further expected to increase with increasing serotiny. It was recently proposed that these high female costs may select for higher hydraulic efficiency in less ramified females (*Harris & Pannell, 2010*). However, this has yet to be directly tested, and the limited available data have found either no difference in hydraulic conductivity between sexes or males with higher conductivity (*Jacobsen et al., 2007*), and no difference in water use efficiency between males and females (*Midgley, 2010*).

We compared two co-occurring *Leucadendron* species: *L. daphnoides* is monomorphic with males and females that have similar degrees of ramification and leaf size, while *L. rubrum* is extremely dimorphic with males being more highly ramified with smaller leaves than females (Fig. 1). First, we developed a metric for quantifying branch ramification that is applicable beyond just *Leucadendron* and show that this metric of ramification scales predictably with traits linked to Corner's rules. Second, we measured whole-shoot hydraulic conductance for shoots ranging in size for all four species-by-sex combinations.

Our analyses showed that more ramified branches (either between sexes or between species) were no less hydraulically efficient, challenging the assumption that higher ramification results in a lower hydraulic efficiency.

## METHODS

### Plant material

Males and females of both species grow naturally near Du Toitskloof Pass, Western Cape, South Africa. All individuals were co-occurring within ∼20 m of each other on a slope extending westward toward the city of Paarl. The two species differ in the degree of serotiny, with *L. rubrum* holding its cones for an average of 2.8 years and *L. daphnoides* for less than one year (*Williams, 1972*; *Harris & Pannell, 2010*). Plants were collected and measurements made during November–December 2012 and during April–May 2013. In the field, mature plants were chosen, ignoring any unusually small individuals. Shoots were cut at the plant base with garden shears and immediately recut under water one node above the previous cut. Individual shoots were placed in dark, plastic bags, and their bases submerged in water during transport back to the lab. Shoots were stored in a 4 °C refrigerator until the day of measurement, and any shoots not measured within three days of field collection were discarded.

### Measurement of hydraulic conductance

Immediately prior to hydraulic measurements, shoots were defoliated underwater. When necessary, a sharp blade was used to cut leaves at the petiole base. Stems were recut underwater with a new razor blade. Hydraulic conductance was measured on whole, branching shoots of various sizes using a low pressure flow meter (*Kolb, Sperry & Lamont, 1996*). With this method, the cut stem base was inserted into a compression fitting (Omnifit) that was connected to hard-sided tubing, the opposite end of which was submerged in a vial of filtered 0.01 M KCl that sat on a balance (Mettler-Toledo MS205DU with a resolution of 0.01 mg). The branching shoot was then placed inside a chamber connected to a vacuum pump. Flow rates from the balance were recorded automatically every 5–20 s (the frequency depended on the absolute flow rate) at each of a series of pressures below ambient: 10, 30, 50, 65, 40, 20 kPa. The stable flow rate at each pressure was determined when the coefficient of variation of the previous ten instantaneous flow rates was less than 5%. Hydraulic conductance, $K$, was obtained by a linear regression of flow rate as a function of pressure (kg s$^{-1}$ MPa$^{-1}$). To compare hydraulic efficiency between groups, $K$ was normalized by either entire shoot leaf area ($K_{LA}$) or by cross-sectional area of the stem base ($K_{CSA}$).

### Quantifying ramification and other traits

At the end of each measurement, shoots were assessed for other morphological traits. The cross-sectional area of the stem base was calculated from the average of two perpendicular measurements of stem diameter at the base using manual calipers. Shoot ramification was quantified in two ways. First, we quantified the rate of diameter change down the length the shoot, using a previously described method *Harris & Pannell (2010)* and described

briefly here. Starting at the highest branch, we measured stem cross-sectional area at the midpoint of each consecutive internode down the shoot as well as the relative position of this midpoint along the length of the shoot. The slope of the relationship between the logarithm of stem cross-sectional area and the relative distance from the crown provides an index of ramification. In the previous application of this method, the tallest shoot on the plant was used and all nodes from the top t the base of the plant were included. However, because we sought to directly link shoot reamification to hydraulic conductance and wanted to capture a wide range of shoot sizes, the shoots we measured were not as large as those measured previously by *Harris & Pannell (2010)*. Nonetheless, because their method for quantifying ramification depends on a linear regression, removing the basalmost nodes should not unduly impact estimates of ramification so long as the assumptions of linear regression are met. Furthermore, the method of quantifying ramification used by *Harris & Pannell (2010)* assumes that all branches arise immediately below terminal inflorescences and at nodes. For some species of *Leucadendron*, including *L. daphnoides*, this assumption is valid. However, in other species, particularly males of *L. rubrum*, most branches occur along the internodes. To account for these many small branches we measured the total number of branch tips on a shoot (regardless of their size and position on the shoot) normalized by the cross-sectional area of the stem base. The number of branch tips per stem cross-sectional area (BTSA) can easily be applied to shoots of any size, including shoot segments from the most recent year of growth that do not have properly defined internodes, as well as to other species with different branching patterns.

Leaves were placed on a flatbed scanner and their total area and number determined per shoot using ImageJ (*Schneider, Rasband & Eliceiri, 2012*). The average leaf size per shoot (termed here simply 'leaf size') was determined by dividing the total leaf area by the total number of leaves on the shoot. For consistency with Corner's rules, we quantified the average leaf area per branch by dividing the total shoot leaf area by the total number of terminal branches on each shoot.

## Data analysis

All data were analyzed using R v. 3.0.2 (*Wickham, 2017*; *R Core Team, 2018*). Scaling relationships were determined using standard major axis regression (SMA) as implemented in the 'smatr' package (*Warton et al., 2012*). For each pair of variables, relationships were determined for each species and sex combination, and differences between groups in scaling relationships determined using a likelihood ratio test as implemented in the *sma* function. For certain scaling relationships, slope tests were used to test whether slopes significantly differed from unity. If the likelihood ratio test (LRT) revealed no significant differences in scaling slopes between groups, data were pooled and the relevant slopes and statistics shown and reported. We also compared $K_{LA}$ and $K_{CSA}$ between groups using ANOVA with species, sex, and the interaction between them as factors. If either of these grouping variables were significant, post-hoc Tukey HSD tests were used to determine which groups were significantly different. For visualization purposes, 95% confidence intervals around scaling slopes were estimated by bootstrap sampling 1,000 iterations.
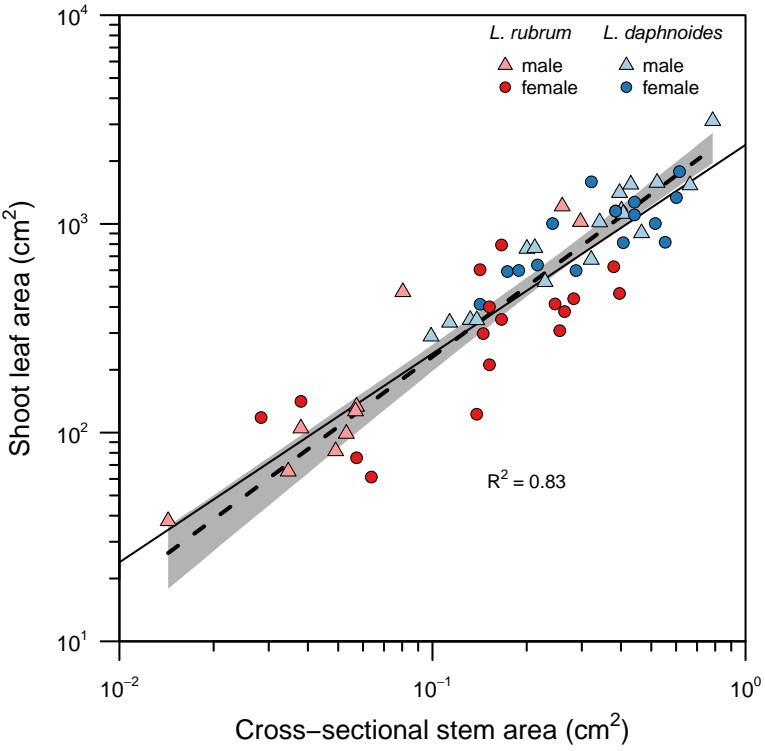

**Figure 2** **Relationship between total shoot leaf area and stem cross-sectional area.** A single standard major axis regression described the scaling relationship across species and sexes and the slope did not significantly differ from unity.

## RESULTS

### Stem and leaf architectural relationships

Stems with larger basal cross-sectional areas held more leaf area (Fig. 2). There was no significant difference in slopes between sex or species groups (LRT = 4.146, $df = 3$, $P = 0.246$), and the global slope was significantly greater than unity (slope = 1.11, $R^2 = 0.83$, $P < 0.001$).

The two metrics of branch ramification did not covary linearly, even in log–log space (Fig. S1.). Subsequent analyses focused solely on BTSA because it can be readily applied to any branching architecture and it conformed to Corner's rules (Fig. 3). More highly ramified branches held significantly less leaf area per branch tip (slope $= -0.97$, $R^2 = 0.93$, $P < 0.001$; slope test: $r = -0.10$, $df = 65$, $P = 0.41$) with no significant difference between groups (LRT $= 1.50$, $df = 3$, $P = 0.68$; Fig. 3A). Similarly, more highly ramified shoots had smaller leaves on average (Fig. 3B). Although species exhibited different elevations in their relationships, there was no significant difference in the slopes among groups (LRT $= 2.74$, $df = 3$, $P = 0.43$; *L. rubrum*: slope $= -0.39$, $R^2 = 0.87$, $P < 0.001$; *L. daphnoides*: slope $= -0.45$, $R^2 = 0.46$, $P < 0.001$).

To determine whether larger branches were more hydraulically efficient, we tested whether there were different scaling slopes between $K$ and two metrics of shoot size: total
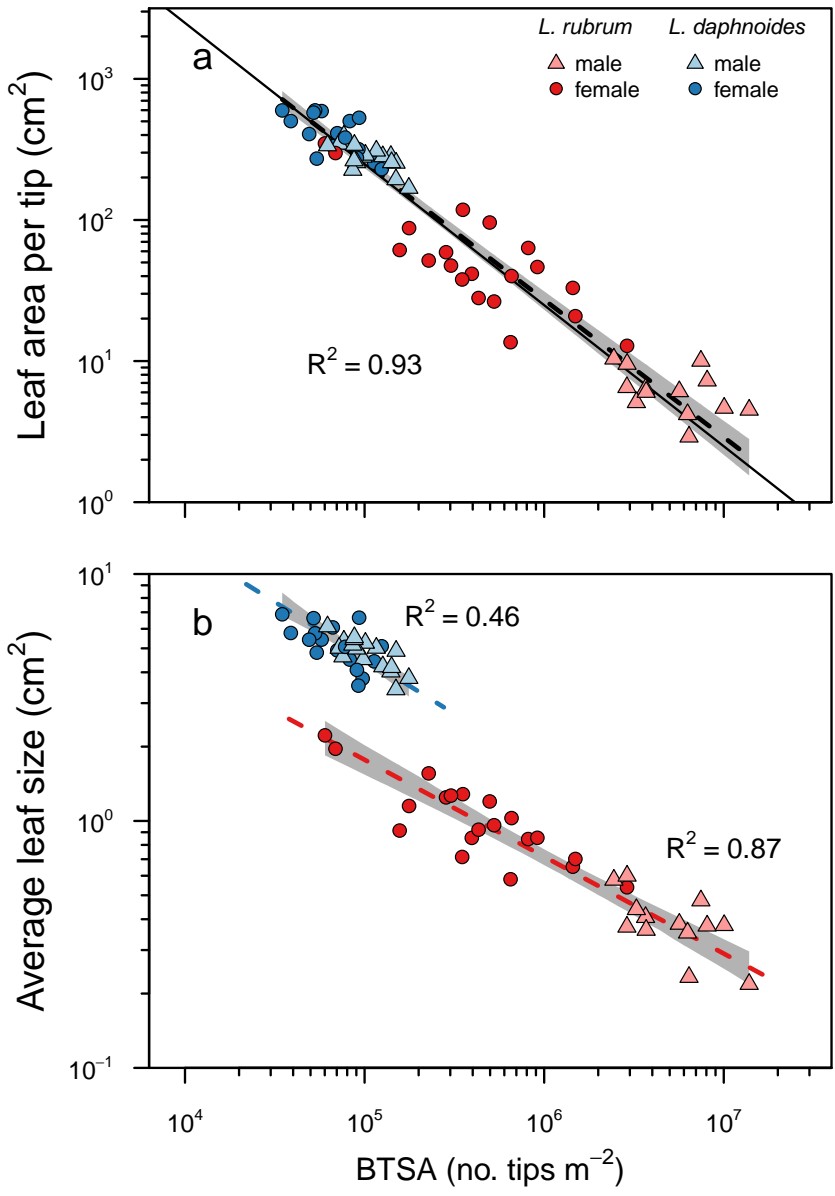

**Figure 3** **Corner's rules and the relationships between branch ramification (BTSA), leaf size, and leaf area per branch tip.** Solid lines indicate 1:1 relationship, and dashed lines indicate the standard major axis regressions, with black lines indicating across all data and colored lines indicating only within specific groups. (A) Relationship between leaf area per tip and BTSA was explained by a single scaling relationship, whose slope was not significantly different from isometry. (B) Relationship between average leaf size and BTSA was highly significant within each species, and the two slopes were not significantly different from each other.

leaf area and basal cross-sectional area (Fig. 4). There was no significant difference in slopes between groups for the scaling of $K$ and leaf area (LRT = 6.19, $df = 3$, $P = 0.10$), and the slope test revealed that the scaling slope was not significantly different from unity ($r = 0.0097$, $df = 66$, $P = 0.94$; Fig. 4A). Thus, total shoot leaf area was a strong

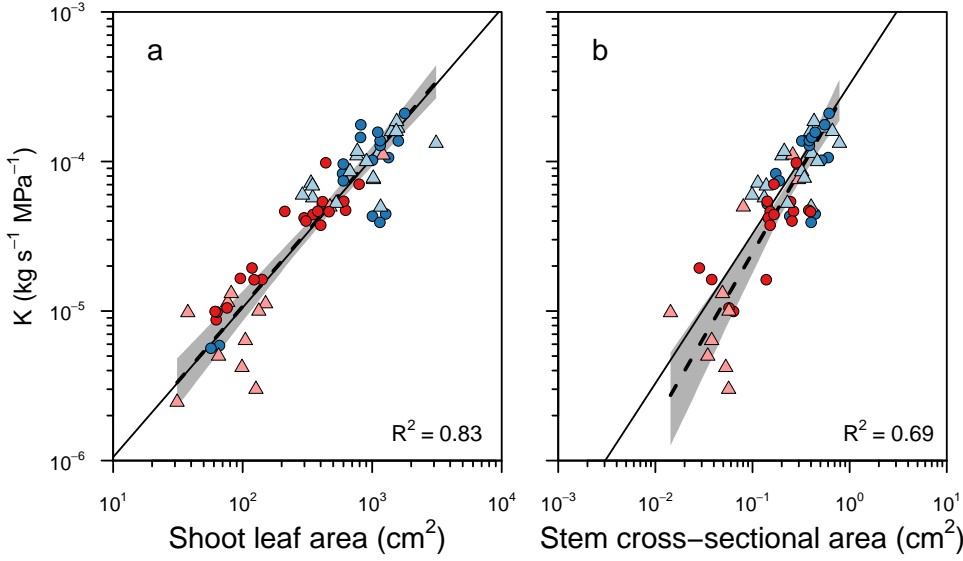

**Figure 4** **Relationships between whole shoot hydraulic conductance and shoot size.** Relationships between whole shoot hydraulic conductance ($K$) and (A) shoot leaf area and (B) cross-sectional area of the stem base. There was no significant difference between sexes or species in these scaling slopes, and a single scaling relationship applied to all data. Grey shading indicated 95% confidence intervals around the slope of the relationship. Solid lines indicate the 1:1 relationship, and dashed lines indicate the standard major axis regression.

predictor of whole-shoot hydraulic conductance across species and sexes ($R^2 = 0.83$, $P < 0.001$). Similarly, there was no significant difference between groups in the scaling slopes between $K$ and stem cross-sectional area (LRT = 5.93, $df = 3$, $P = 0.12$), and there was no significant difference between the observed slope and unity ($r = 0.212$, $df = 58$, $P = 0.11$; Fig. 4B). Thus, basal cross-sectional area was a strong predictor of whole shoot hydraulic conductance across species and sexes ($R^2 = 0.69$, $P < 0.001$).

To determine whether ramification impacted hydraulic efficiency, we tested whether there was a significant relationship between BTSA and leaf area-specific hydraulic conductance ($K_{LA}$) and between BTSA and hydraulic conductance ($K$) normalized by stem basal cross-sectional ($K_{CSA}$). There was no universal effect of BTSA on either $K_{CSA}$ ($P = 0.79$; Fig. 5A) or $K_{LA}$ ($P = 0.13$, Fig. 5B). Within certain species and sex groups, however, there were some significant effects of BTSA. Only in *L. daphnoides* females did more highly ramified individuals have lower $K_{CSA}$ ($R^2 = 0.30$, $P = 0.02$; Fig. 5A). In contrast, more ramified *L. rubrum* males had higher $K_{CSA}$ ($R^2 = 0.42$, $P < 0.02$; Fig. 5A). There was no significant effect of species, sex, or the interaction between species and sex on $K_{CSA}$ in an ANOVA (all $P > 0.05$). Similarly, only for *L. daphnoides* females was $K_{LA}$ lower for more ramified individuals ($R^2 = 0.36$, $P = 0.01$; Fig. 5B), and when *L. rubrum* males and females were pooled together there was a weak, but significant negative relationship between BTSA and $K_{LA}$ ($R^2 = 0.16$, $P = 0.02$; Fig. 5B). ANOVA revealed a similar pattern, with a significant effect of the interaction of species and sex ($F = 4.59$, $df = 1$, $P = 0.04$) on $K_{LA}$. However, a Tukey post-hoc HSD test found no single pairwise comparison significant,

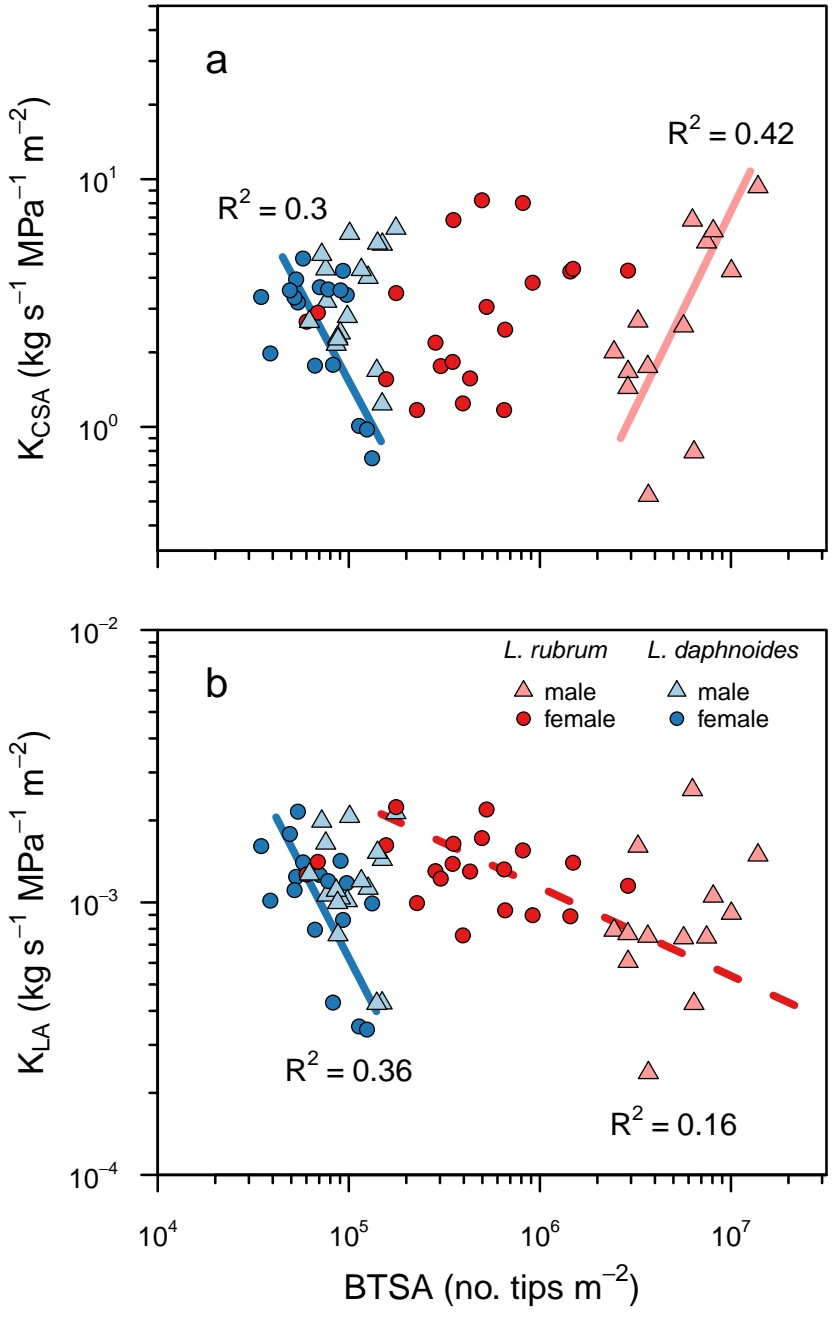

**Figure 5** **Shoot ramification had little impact on shoot hydraulic efficiency.** (A) Shoot hydraulic conductance per unit cross-sectional stem area as a function of BTSA. (C) Shoot hydraulic conductance per unit leaf area as a function of BTSA. Solid colored lines indicates a significant relationship within a sex × species combination, and dashed colored lines indicate a significant relationship within a species (*P* < 0.01).

even before Bonferroni adjustment, at the $P = 0.05$ level. The largest pairwise difference was between *L. rubrum* males and females, although this difference was not significant in either the Tukey post-hoc comparison ($P = 0.14$) or in a Welch's two-sample $t$-test ($t = 1.997$, $df = 18.55$, $P = 0.06$).

## DISCUSSION

In the present analysis of two co-occurring *Leucadendron* species differing in their degrees of sexual dimorphism, we found no consistent support for hydraulic differences between the sexes, or even between species. Despite substantial differences in gross morphology, there was little effect of variation in leaf or stem morphology on hydraulic conductance and hydraulic efficiency. Overall, these results suggest that while Corner's rules dictate the relationships between leaf size, inflorescence size and number, and branch ramification, there is little net effect of these traits on the hydraulic functioning of the stem. Previous work has argued that in *Leucadendron* the accumulation of resistance in branch junctions and the requirement for serotinous females to supply water to long-lived seed cones would drive sexual dimorphism in branch ramification (*Harris & Pannell, 2010*), but our results do not support the underlying assumptions of these ideas.

### Allometry of leaves and stems obeys Corner's rules

Corner's rules predict that "the stouter the stem, the bigger the leaves and the more complicated their form" (*Corner, 1949*). Corner's first prediction between stem size and leaf size has been well-characterized for many species including *Leucadendron* (*Bond & Midgley, 1988*; *Ackerly & Donoghue, 1998*; *Olson, Aguirre-Hernández & Rosell, 2009*). Across species and sexes a single slope explained the relationship between total shoot leaf area and stem cross-sectional area with larger stems holding proportionally more leaf area (i.e., SMA slope not significantly different from unity; Fig. 2). This relationship further suggests that despite large variation in stem diameter both within and among species, leaf area is allocated proportional to stem diameter.

More apropos to the present study, *Corner (1949)* also predicted that "the greater the ramification, the smaller become the branches and their appendages". This prediction emerges out of an argument of allocation: given a constant amount of biomass, producing more branches means that each of them must be smaller. In most species of *Leucadendron*, branches emerge just below terminal inflorescences and seed cones. This branching pattern results in internodal segments bracketed by nodes from which a variable number of branches emerge. In dimorphic Leucadendron, males have more branches at each node than their conspecific females, which is driven by pollinator selection for more inflorescences because inflorescences are borne only on terminal branches (*Bond & Maze, 1999*). Selection on inflorescence number, therefore, can result in more highly ramified branches and, because of Corner's rules, smaller leaves (*Midgley & Bond, 1989*). Using BTSA as a simple metric of ramification, we found that more highly ramified shoots both within and among species and sexes bore less leaf area per branch, driven predominantly by each leaf being smaller (Fig. 3). These results were consistent with Corner's second prediction, with isometric scaling between BTSA and leaf area per branch tip. Interestingly,

while leaves were smaller on more ramified shoots, there was a species-specific effect such that the two species had different intercepts (Fig. 3B). In contrast, another metric of ramification recently used in a broad survey of *Leucadendron* species (*Harris & Pannell, 2010*) exhibited no similar scaling relationships with leaf size or with leaf area per tip (Fig. S2).

## Little effect of shoot morphology on shoot hydraulics

We sought to test whether these large differences in leaves and stems among species and sexes in *Leucadendron* led to differences in stem hydraulic efficiency, as has been suggested (*Harris & Pannell, 2010*). Two main characteristics of woody shoots were thought to be important: the cross-sectional area of a branch and the number of branching nodes. These two traits are, to some extent, linked in *Leucadendron* because more highly ramified shoots are more likely to have more nodes and smaller terminal branches. As a rough approximation, larger diameter branches should have higher conductance because they have more xylem conduits (assuming no change in conduit size with branch size). However, wood is complex, being composed of vessels, fibers, parenchyma, and pith in various proportions (*Zanne et al., 2010*), such that hydraulic conductance does not necessarily scale isometrically with cross-sectional area. The complexity of wood structure-function relationships means that simplistic models that treat the stem as a pipe cannot accurately describe shoot water transport capacity (*Lehnebach et al., 2018*). For example, because stems continually grow, their theoretical hydraulic capacity always increases, yet as xylem ages it becomes progressively non-functional, such that most of the water transpired by leaves is delivered via the current year's xylem (*Melcher, Zwieniecki & Holbrook, 2003*; *Brodersen et al., 2019*). As a result, cross-sectional area of the stem cannot necessarily predict the hydraulic capacity of the stem.

Despite numerous measurements on the hydraulic conductivity of stems, surprisingly few studies have examined the hydraulic conductance or conductivity of branch junctions (nodes). The available data suggest that branch junctions increase hydraulic resistance, but the magnitude of this effect is highly species specific (*Ewers & Zimmermann, 1984b*; *Ewers & Zimmermann, 1984a*; *Ewers, Fisher & Chiu, 1989*; *Tyree & Ewers, 1991*; *Tyree & Alexander, 1993*). In *Leucadendron*, most species produce new branches each year almost exclusively just below terminal inflorescences and, in the case of females, seed cones. Leaves are borne on newer branches, and so transpired water must traverse multiple nodal junctions before it reaches the leaves. Yet, because the effect of junctions on hydraulic resistance is species-specific, simply having more nodes does not necessarily translate into higher resistance.

We found no compelling evidence that hydraulic efficiency differs between conspecific males and females or even between the co-occurring species we studied here, which differ dramatically in their degrees of sexual dimorphism. At the entire shoot level, ramification had little impact on hydraulic efficiency, whether determined on a leaf area basis or on a stem cross-sectional area basis (Fig. 5). Among *L. daphnoides* females more highly ramified individuals had lower $K_{CSA}$, but *L. rubrum* males–which were the most highly ramified shoots we measured–exhibited the opposite relationship. When measured on a leaf area

basis, hydraulic efficiency was lower in more highly ramified *L. daphnoides* females and among *L. rubrum*, but only when pooling males and females together. Overall, these results suggest that ramification *per se* has little impact on hydraulic efficiency at the level of the entire shoot. While branch junctions may add resistance to the shoot, the amount of resistance is likely small and compensated for by other adjustments, such as changes in leaf area or additional xylem area produced in basal segments. Furthermore, assuming a given length of stem growth, adding this stem as an additional branch rather than by growing the length of an existing branch would actually be beneficial because resistances are additive in series and conductances are additive in parallel. Thus, adding a stem length as a branch in parallel would add to the conductance of the entire shoot rather than subtract from it, as adding that stem length in series would. This feature of resistance networks could compensate for the resistance of adding a branch node, and other studies have suggested that trees may become more ramified as they grow taller (i.e., they add parallel conductors) in order to compensate for the added resistance a longer path length (*Mencuccini & Grace, 1996*). The similar water use efficiencies of *Leucadendron* males and females supports the equivalence of their hydraulic efficiencies (*Midgley, 2010*). Thus, larger stems, whether males or females, provide no more water to their transpiring leaves per unit leaf area or stem cross-sectional area than do smaller stems, in contrast to some arguments about how differences in hydraulic efficiency between males and females may be linked to sexual dimorphism in the genus (*Harris & Pannell, 2010*).

Males and females of each species must survive to the next fire event in order to be represented in the most recent year's seed crop and, thus, maximize fitness (*Midgley, 2000*). Conspecific males and females should, therefore, have similar longevities and likely also similar rates of metabolism. This is supported by the similarity in hydraulic efficiencies among conspecifics we studied and the strong correspondence between shoot hydraulic efficiency and photosynthetic capacity (*Brodribb & Feild, 2000*; *Brodribb, Holbrook & Gutiérrez, 2002*). If the hydraulic architecture of females were more efficient than that of males, then it is not clear why males would not also have evolved a similar branching and leaf structure given that males and females co-occur and may compete with each other. Also, if the hydraulic architecture of males were less efficient, then males may incur costs associated with reproduction equal to or higher than those of females, contrary to most theory and data regarding the costs of reproduction (*Bond & Maze, 1999*). Instead, our data and those of *Midgley (2010)* are consistent with there being similar hydraulic efficiencies between species and sexes, regardless of the degrees of sexual dimorphism and branch ramification.

## CONCLUSIONS

Despite substantial variation between species and sexes in morphological traits of leaves and stems, these differences did not translate into differences in hydraulic efficiency. More highly ramified individuals had smaller leaves, consistent with Corner's rules, but there were no consistent effects of these morphological modifications on hydraulic efficiency. These results suggest that selection may act on males and females in *Leucadendron* to

maintain physiological performance while allowing for large variation in morphological traits that could be under divergent selection. These results further highlight the importance of recasting studies of hydraulic efficiency at the level of the entire shoot by quantifying the pathways of water transport all the way to the leaves.

## ACKNOWLEDGEMENTS

RP Skelton assisted in sample collection in the field.

### Funding

Adam Roddy was supported by a U.S. National Science Foundation Graduate Research Fellowship. The funders had no role in study design, data collection and analysis, decision to publish, or preparation of the manuscript.

### Grant Disclosures

The following grant information was disclosed by the authors:
U.S. National Science Foundation Graduate Research Fellowship.

### Competing Interests

The authors declare there are no competing interests.

### Author Contributions

- Adam B. Roddy conceived and designed the experiments, performed the experiments, analyzed the data, contributed reagents/materials/analysis tools, prepared figures and/or tables, authored or reviewed drafts of the paper, approved the final draft.
- Justin J. van Blerk performed the experiments, analyzed the data, contributed reagents/materials/analysis tools, authored or reviewed drafts of the paper, approved the final draft.
- Jeremy J. Midgley conceived and designed the experiments, authored or reviewed drafts of the paper, approved the final draft.
- Adam G. West conceived and designed the experiments, contributed reagents/materials/analysis tools, authored or reviewed drafts of the paper, approved the final draft.

### Data Availability

The raw data is available in the Supplemental File.

### Supplemental Information

Supplemental information for this article can be found online at http://dx.doi.org/10.7717/peerj.6835#supplemental-information.

# PeerJ

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
