# Peer review of "Ramification has little impact on shoot hydraulic efficiency in the sexually dimorphic genus Leucadendron (Proteaceae)"

_PeerJ, doi:10.7717/peerj.6835_

## Round 0.1 · original submission · Major Revisions

The reviewers' comments have come in, and two have suggested accept with revisions, whereas the author who had their work critiqued suggests reject. By addressing the following comment your results could be published without the emphasis on whether or not Pannell was correct. In fact, the question of interest is whether plant branching, can explain hydraulic traits, rather whether a previous scientist was correct or not.

Firstly, reword the text to re-focus the question which is most important (e.g., are their important links between specific-conductive capacity and branch ramification), rather than concentrating so exclusively on one particular study, i.e., Harris and Pannell 2000. One Reviewer comments that this has the effect of putting Harris and Pannell on the defensive, and detracts from the main results, i.e., it is probably not so important that Harris and Pannell were wrong, but that hydraulic constraints do not explain branching ramification more generally.

Second, please clearly and concisely address the concerns by Pannell. I suggest removing the method called as 'HP' and state something like 'previous methods'. Please address the comments he made on branch collection and size of branches sampled.

Please avoid citing honours theses.

Also please clarify the K_LA and K_CSA data - two reviewers comment on Figure 5 and that leaf area and/or cross sectional area are used in K_LA, and thus, please comment on where auto-correlation occurs. If auto-correlation occurs, and this is not important - then state this. If you want to avoid presenting the auto-correlation then this needs addressing. I suggest removing auto-correlation where possible unless absolutely necessary as it weakens the credibility of the entire paper.

·

Basic reporting

No comment; see pdf

Experimental design

No comment; see pdf

Validity of the findings

No comment; see pdf

Additional comments

No comment; see pdf

Reviewer 2 ·

Basic reporting

Clear language and generally well-written. Most of the relevant literature has been cited. The manuscript has a professional structure (but see my general comments below).

Experimental design

Very clear experimental design. Well thought out design and implementation. The research question is well defined and mostly well-laid out. Rigorous investigation performed to a high technical standard. Methods well described.

Validity of the findings

The results are conclusive, despite being negative (see my comment below). Conclusion is well stated and links strongly to the original question. I have queried the relevance of one or two figures (see my comments below), but mostly the figures and appropriate.

Additional comments

The study “Hydraulic constraints of reproduction do not explain sexual dimorphism in the genus Leucadendron (Proteaceae)” by Roddy et al. investigates the potential causes of sexual dimorphism in dioecious species of Leucadendron. Specifically, Roddy et al. test the hypothesis that sexual dimorphism in branching patterns is caused by differing hydraulic demand to reproductive structures. The authors pick up on an earlier paper by Harris and Pannell (2010) suggesting that, because female cones require greater hydraulic supply than male cones in Leucadendron, female branches should be less branched to improve “hydraulic efficiency”. To do so Roddy et al. examine the extent of branching in males and females of two Leucadendron species (L. rubrum and L. daphnoides) and compare this to measures of hydraulic conductance. Although the authors find some evidence of an association between hydraulic conductance and branching patterns in L. rubrum and female L. daphnoides (but not males), they conclude that there is no clear relationship between branching and “hydraulic efficiency” in the genus.
I think that the study is well thought out and thorough in its scope. I agree that there appears to be no clear trend between branching (ramification) and hydraulic conductance, and that alternative explanations for dimorphism should be assessed. Even though the results from the study are negative (i.e. the authors find little evidence to support the Harris and Pannel hypothesis), the science is good and the results are interesting.
My biggest issue with the manuscript is that the authors take some time to develop their case, and this makes it hard to follow at times. This extends from the introduction (see my comment below in the specific comments section) to the results/figures (see my comment below). I wonder if the authors might make the manuscript simpler to read/follow by dropping the discussion of trade-offs and/or scaling relationships and focussing more on simpler tests of the hypothesis (e.g. comparisons between males and females)? We only start to see the hypothesis being tested in Figure 5 and Figure 6, and I wonder if the authors can find a way of getting there quicker (see my suggestions below).
Specific comments:
Introduction
Line 78-82: What is the journal policy on citing Honours theses?
Line 90-93: The relevance of the cell size paper to the current study is not entirely clear.
Line 112-116: The relevance of this sentence in the context of the introduction is not entirely clear.
At times the introduction comes across as being quite convoluted, jumping from one aspect of the study to another. For example, the introduction moves from introducing dioecy and its potential impacts on plant hydraulic architecture (very clear, well laid-out), to mentioning trade-offs (and predictions for different levels of analysis), to an attempt to define efficiency, and back to trade-offs. I understand that the authors have tried to be incredibly thorough in their assessment (including developing a better metric of branching), and that this required introducing Corner’s scaling laws to defend their metric etc. However, I recommend that the authors shorten the introduction, sticking more to the point of testing whether there is a relationship between plant architecture and hydraulic conductance.
Results
Line 254-264: This is the real heart of the results section and the authors should consider getting to it earlier (and expanding on it). Do males and females differ in hydraulic conductance?
Generally, I feel that the results section could be shortened or laid out slightly differently. For example, the authors use three figures (Figures 2 through 4) to introduce their trade-off framework, to compare ramification metrics, and to introduce scaling relationships. Some of this information could very well be supplementary material. (e.g. Figure 3 could very well be a supplementary figure. If the authors established a better indication of branching then that presented by Harris and Pannell (2010), then mention this in the methods, and move on to test the interesting ideas.)
Figure 5: Since leaf area is incorporated into the metric of kLA, are these variables not autocorrelated?
Figure 6 is clear. Did the authors consider conducting a simple t-test to establish whether k varied between males and females? Simply showing that k is not related to sex would go some way to establishing whether “hydraulic efficiency” is linked to reproductive investment.
Also, why such a large range in k? Two orders of magnitude seems fairly large. Can the authors comment on this?
Discussion
Nice conclusion.

·

Basic reporting

meets standards (please see comments to author)

Experimental design

meets standards (please see comments to author)

Validity of the findings

meets standards (please see comments to author)

Additional comments

General comments

The hydraulic consequences of branch ramification, resulting from sexual dimorphism or otherwise, is an interesting and poorly understood topic. Also, the causes and consequences of sexual dimorphism is also poorly understood in may respects. As such, I think these data are important and likely to be cited in the future. The experimental methods appear to be appropriate and well implemented (very nice in fact). I have two criticisms, which I think can be easily addressed, and are likely not serious flaws providing the authors can indeed address at least the second criticism.

Firstly, I would suggest the authors consider concentrating on the question at hand (e.g., are their important links between specific-conductive capacity and branch ramification), rather than concentrating so exclusively on one particular study, i.e., Harris and Pannell 2000. Not only does this have the effect of putting Harris and Pannell on “front street”, but I think it detracts from the main results, i.e., it is probably not so important that Harris and Pannell were wrong, but that hydraulic constraints do not explain branching ramification more generally. I think most readers would agree with this, and could be address quite easily with just a bit of word crafting.

Secondly, and more importantly, it appears as if the x and y axes in Figures 5a, 5d, and 6a are not independent. In a way, this is not so important for two reasons. Foremost, in the face of this non-independence, the authors’ hypothesis are still upheld. For example in Figure 6a, we should expect positive correlation to result from non-independence, but the authors report none, thus supporting their hypothesis that hydraulics do not limit ramification. Secondly, figures 5a and 5d do not appear to be very important to the hypotheses and it seems to be the differences between the groups (male vs female, and between species) that the authors are interested in, rather than within-group r2 and p values, that are important to the questions being asked. Nevertheless, unless I’ve misunderstood something (and the axes are indeed independent), reporting r2 and p values from non-independent axes is invalid.

I might suggest to the authors that if they think the relationships reported in these figures are still important to their story, to then use standard major axis regression to address them directly. For example, if the question in Fig 5a is, “does shoot shoot conductance scale differently with leaf area between male and female plants (or between the two species), then they could plot conductance on the y axis (Q/pressure gradient) and leaf area on the x axis. Male shoots and female shoots should be plotted individually and the group-specific slopes and intercepts tested using the “sma” function in the “smatr” package.


Specific comments

91: It’s not clear what the independent variables are in this sentence, e.g., is any trait~hydraulic correlation expected to be significant just because there are constraints on cell expansion? Also, it reads as if the authors expect that these constraints will only influence variation on the x axis (hydraulic efficiency). This would not result in covariation between x and y nor consistent coefficients between male and female groups if plotted individually. Perhaps just a bit more elaboration here would help clarify this point.

94-116: I have a few suggestions just to make the arguments and terms clearer in this paragraph. Considering that “efficiency” is indeed such a slippery term (whereas conductance has well understood units and meaning), I wonder if it might be better to simply use something like “mass-specific conductance”… or “volume-specific conductance” to give clear and intuitive meaning to the measurement. Also, I don’t think it is correct to say that conductance divided by “size” (mass?, volume?, height?) is a good measurement of “flux of water per investment cost of moving that water”. After all, water moves through xylem via a “free energy” gradient (energy is provided by the sun/atmosphere). Perhaps it might be more correct to simply state that conductance needs to be scaled by a meaningful metric… and it matters what metric we use (i.e., the authors comparison of mass vs size [volume?]). Or perhaps the authors really mean to normalize k by “the cost of building and maintaining the network”, in which case volume or mass would be more appropriate.

132-133: In the case of positive correlation, I suppose many “hard tradeoffs” could be erroneously dismissed simply because the magnitude of error often correlates with the magnitude of the measurement. This would result in predictably different coefficients for the quartiles. ...just to play Devil’s advocate…

144-145: There is a mistake in the wording. If the conductance has been divided by “cross-sectional area” (144), then it will almost certainly scale inversely with cross-sectional area (145).

166: How many days between collection and the day of measurement?

230-231: I might suggest replacing “it” with “HB ramification” in this sentence, just to be clear. I can’t really compare Fig 3 with table 1 because male and female plants are not denoted in the column headings of table 1. Are there p values to report for the male vs female comparisons?

241-241: I might agree that poor correlation and divergent coefficients among quantiles are indication for an absence of a tradeoff, but I don’t agree that strong correlation and convergent coefficients are indication that correlations are “likley hard tradeoffs”. I think the authors could correctly say that the evidence is congruent with a hard tradeoff (or similar), but not that x and y are linked via cause and effect.

244-253: I am a bit confused by the statistics here. K_LA is K divided by shoot leaf area, right? And K_CSA is K divided by shoot cross sectional area, right? If this is not correct, please clarify in the methods. If this is correct, we should expect negative correlation between these traits simply because the x an y axes are not independent. For example, in Fig 5a, the exact same variation in shoot leaf area (x) also appears in the denominator of the y axis. As such, R2 and P values from the within-group correlations are meaningless and should probably not be reported. HOWEVER, the p values from the across-group comparisons, i.e., comparing first the slope then the intercept (if slopes are similar) using the “sma” function in smatr should still be valid. Considering that these between group comparisons are the point of the manuscript, I don’t see the non-independent axes as necessarily being a major problem… but I would strongly recommend not reporting r2 or p values from auto-correlated axes, and instead report the differences in slope (and intercept if appropriate) and p values of the between group comparisons.

There appears to be a similar problem in figure 6a. For example, I think the cross sectional area appears in the denominators of both the x and y axes. As such, we should expect positive correlation between x and y simply as an artifact of this. However, the authors would still be correct in pointing out that even though we should expect positive correlation between these traits, they still did not, and thus, there appears to be no strong evidence that more highly ramified branches require greater K_csa… which is consistent with the author’s hypothesis. Figures 6b, c, and d appear okay though.

299-300: Again, I think the authors could say that it is not inconsistent with a true tradeoff, but I don’t think this is good evidence that one exists.

Table 1: It is not clear which values are male and which are female from the column headings. Also, the table is missing a bottom line. Would it be possible to report p values for the male-female comparisons?


I would like to thank the authors and editor for giving me the opportunity to review this very interesting study. I sincerely hope that the authors find at least some of my comments useful.

Sean M. Gleason

---

## Round 0.2 · accepted · Accept

Please re-write L282 to 287 to remove the highlight on the method belonging to "Harris and Pannell". Please replace with text like 'we quantified the rate of diameter change... using a previously described method (Harris and Pannell, 2010)".

Also 'In using this method to quantify ramification, the tallest shoot on a plant was measured and nodes all the way..., were measured'.

The importance of the work is progressing or evolving a a method, not including the name of the person. Given the comments from reviewers I suggest removing attention from the person and focus on the methods.

Minor edits:
The font changes from abstract to the rest of the paper and headings. Please make consistent.
Please rephrase first sentence of abstract:
“Despite the …… the impact of this morphological variation on hydraulic efficiency has been poorly studied.”
LINE 53: / higlight / ? / [Is this an artifact to be removed?]
LINE 69: / - or . . . segment - / (or . . . segment) / [parentheses instead of dashes?]

#